# Experimental Verification of the CFD Model of the Squeeze Film Lifting Effect

Bartosz Bastian, Rafał Gawarkiewicz, Michał Wasilczuk * and Michał Wodtke

Faculty of Mechanical Engineering and Ship Technology, Gdańsk University of Technology,
Narutowicza St 11/12, 80-233 Gdańsk, Poland; bartosz.bastian@pg.edu.pl (B.B.)
* Correspondence: mwasilcz@pg.edu.pl

**Abstract:** The presented study shows the results of the research into the squeeze film levitation phenomena. The system introduced in the investigation is composed of a vibrating surface, air squeeze film, and the surface of the body freely suspended over the film. The use of the CFD (Computational Fluid Dynamics) model used in the system allows us to determine the steady state, periodic behavior of the air film (described by Navier–Stokes, continuity equations, and ideal gas law), and the lifted object dynamics. The model allows us to determine multiple factors, among others, mean film thickness and pressure distribution inside the fluid film. The influence of factors, such as vibration amplitude, frequency, and load on the lifting conditions, was presented. A series of calculations show the levitations height in the range of 5.61 up to 58.12 microns, obtained for masses of samples between 5–20 g, vibration frequency of 5–25 kHz, and the motions amplitude of 0.5–1.5 μm. A series of CFD multivariable calculations for a standing wave inducer were not previously published. The CFD model was validated with the use of experiments on a specially developed test rig. The authors experimentally obtained the height of levitation up to 200 microns.

**Keywords:** CFD calculations; acoustic levitation; near-field acoustic levitation (NFAL); squeeze film acoustic levitation (SFAL); squeeze film effect

## 1. Introduction

Levitation is a phenomenon when a body is suspended in an environment without any mechanical support. The effect is achieved by arranging a system counteracting a gravity pull. The balancing forces can be induced with the utilization of multiple methods, i.e., magnetic levitation, electrostatic levitation, aerodynamic levitation, acoustic levitation, etc. Two distinct methods can be identified under the term acoustic levitation. Standing-wave acoustic levitation (SWAL) and near-field acoustic levitation (NFAL) are also referred to as squeeze film acoustic levitation. SWAL is based on the effect of the standing wave; the levitating object can be located in the nodes of the acoustic pressure field lying between the acoustic source and the reflector [1]. In the NFAL, the reflector simultaneously plays the role of the suspended object. Moreover, both types of levitation can be identified through the dominant direction of the gas movement in the film. The main gas velocity direction normal to the plane of the inducer is characteristic of the SWAL, whereas, in NFAL, the primal velocity component is parallel to the length of the film, similar to the case of the squeeze film effect.

### 1.1. Applications

One of the major advantages of levitation is the lack of contact between the inducer and the lifted object. The contactless support allows for reducing the wear due to contact and resistance due to friction in comparison to regular bearing systems. For almost three decades, researchers have studied the phenomenon of NFAL [2]. Over the years, multiple possible applications have been proposed and studied with varying configurations.

One of the first works was published by Hashimoto et al. [2]. It shows the possibility of utilizing a vibrating plate as a means of suspending the object above itself. The researchers concluded the conditions of stable levitation. For flexural vibrations, the size of the object should be greater than half the wavelengths of the flexural vibration. Dependency between the height and the amplitude of vibration was also observed. However, the proportionality factors differed from one specimen to the other.

Yoshimoto et al. were one of the first to propose the linear guiding system utilizing the squeeze film effect [3,4]. The system used hinges as a way of generating a deformation. In the system used, applicable frequencies were limited to 3 kHz, resulting in high noise. Improvement of the idea was later presented in [4]. The novel design worked in ultrasonic frequencies. In both cases, the float characteristics were studied to determine the possible application.

The transportation system was also proposed by Ide et al. [5]. The set-up consisted of a plate placed over right-angle beams located in a way that formed a prism-shaped rail (Figure 1). The beams acted as a guiding rail for the plate and also, while vibrating, created a traveling wave, allowing for the plate to move along the rail guides

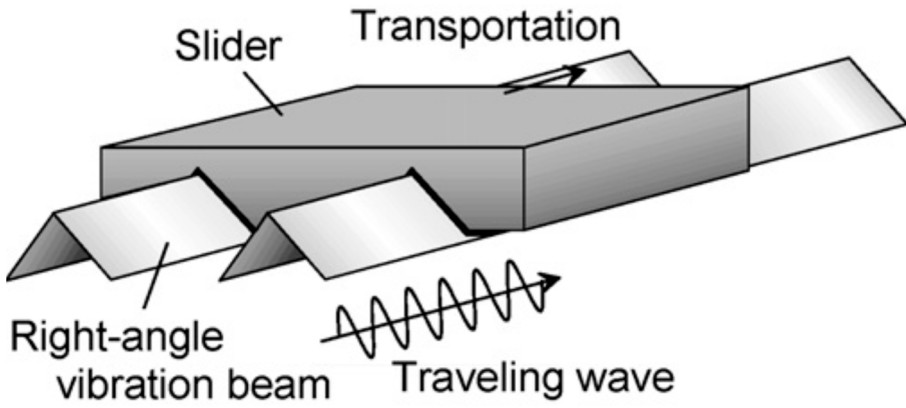

**Figure 1.** The basic idea of a non-contact linear bearing using flexural vibration of a pair of right-angle beams. Reproduced from [5] with permission from Elsevier, © 2007.

An alternative idea for a linear motion system was provided by Bucher et al. [6]. The design utilized the idea of two independent actuators controlled collectively. Actuators would create an inequality of heights between the changing planes, thus creating a deflection angle between the plane of actuators and the sample. This inequality would allow for the tangential motion of the sample.

Another application was proposed by Stolarski et al. [7]. In their work, a journal bearing was proposed. The work focused on finding a proper arrangement of such a bearing. The deforming part was a deformable bush allowing for frictionless rotation of the shaft. Later, operational conditions of such and a new configuration were studied [8,9].

A motor utilizing the NFAL was designed by Bucher et al. [10]. In their work, the stator annulus was deformed by three actuators. The generation of the traveling wave in the ring allowed for the rotation of the disk suspended above the ring.

A spherical bearing proposed by Chen et al. [11] was created as a frictionless suspension for the gyro. The hemispherical bowl acted as a stator, and the sphere mounted atop was suspended and successfully rotated, giving a possibility to use NFAL for suspended gyro design.

*1.2. Experimental Research*

1.2.1. Inducer Type

One of the main distinctions in the NFAL setup that can be made is by differentiating the type of inducer used. The first uses the piston-like generator [12,13], a single, mostly rigid plate with little or no deflection. Alternatively, the use of flexural mode can be

observed. Single or multiple actuators work near the resonant frequency of the inducer. This generally allows for lower energy consumption during the operation as the vibration occurs in the natural frequency of the inducer. On the other hand, generating a forced vibration in a piston-like device, where no means of energy accumulation occurs, seems to require more energy, although the authors do not know of any systematic comparisons of the two methods.

The use of the flexural mode is also highly beneficial because it can be used not only as a suspension but also as a transportation system [5,14,15]. In most cases describing the transportation system application, the traveling wave is used as presented above. The standing wave, similar to the rigid plate solution, is mainly used for its lifting/suspension capacity.

### 1.2.2. Measurement Techniques

In the experimental setups, the most commonly measured characteristic is the height of levitation. The measurement can be easily completed with the use of optical displacement sensors or eddy current sensors [16]. Researchers trying to measure other properties are rare. Stolarski [17] described using the pressure sensor in the setup of a sliding bearing. The measurement showed the average dynamic pressure of the film. Zhao et al. [18] used a force sensor to determine the load-carrying force in the proposed journal bearing. In addition, for studies dealing with the forced motion of the sample, inertial sensors [11] and shaft speed sensors [7] were used.

### 1.3. Unresolved Issues

There is a lack of clear methods to predict the levitation height found in the research. Recalled applications (i.e., bearings,) due to the geometrical limitations, do not deal with the height but rather an acceptable load [7,9].

Calculation models utilized in the papers describing the behavior of the gaseous film are commonly based on Reynolds equations. The equations are derived and simplified Navier–Stokes equation of motion of the fluid. The limitation of using Reynolds in comparison to general motion equations was discussed by Brunetiere et al. [19]. Their work showed that aside from Reynolds, Helmholtz's number should also be considered in choosing the proper model for the film.

The results presented in the articles usually show a levitation height [20,21]. Calculation methods often provide an analysis of pressure distribution in the film [19,22,23]. A few works show the transient motion of the lifted object [12]. In many cases, it is due to the higher inertia of the objects, where the amplitude of the vibrating motion of the sample is negligible in comparison with the levitation height.

In calculations of the NFAL phenomena, the lifted object is usually modeled as a rigid body; while bearing in mind possible applications in transporting vulnerable objects, it is important to be able to assess the deflections and stresses of the lifted object. That is why the authors' goal is to build a full FSI (fluid–structure interaction) model of the phenomena, also because it seems extremely difficult to measure the stresses and deflections of a lifted object in an experiment. The model presented here is a step toward building a full FSI model.

## 2. Materials and Methods

### 2.1. Experimental Research

The idea of the experimental setup and the photograph are presented in Figure 2. The vibrating circular membrane is peripherally fixed. The piezoelectric actuator fixed with a special adhesive beneath the membrane is made from lead zirconium titanate $Pb(Ti_xZr_x)O_3$ [24], commonly referred to as PZT. The alternating current supplied to the PZT element generates vibrations of said membrane. If the current frequency is correlated with the natural frequency of the plate, with sufficient energy supplied, the vibration of the membrane can be observed. Above the working surface, the sample is placed to observe the possible levitation phenomenon resulting from the formation of the gas film due to vibrations of the vibrating plate, as shown in Figure 2c. The measurements of the

movement of either the vibrating plate or the lifted sample are accomplished with the use of the confocal optical sensor IFS2403-0.4 distributed by Micro-Epsilon. The sensor has a 0.4 mm measuring range with a dynamic resolution of 47 nm. The power supply for the PZT element allows for control of the offset voltage and the amplitude of the AC. The power supply was defined as 50 V, with an amplitude of 5, 10, 20, and 40 V. The frequencies generated by the setup are in the range between 1 Hz and 54 kHz. Data were collected using a confocal DT 2471 HS distributed by Micro-Epsilon. The optical sensor readings and power supply voltage were collected with a sampling frequency of 50 kHz.

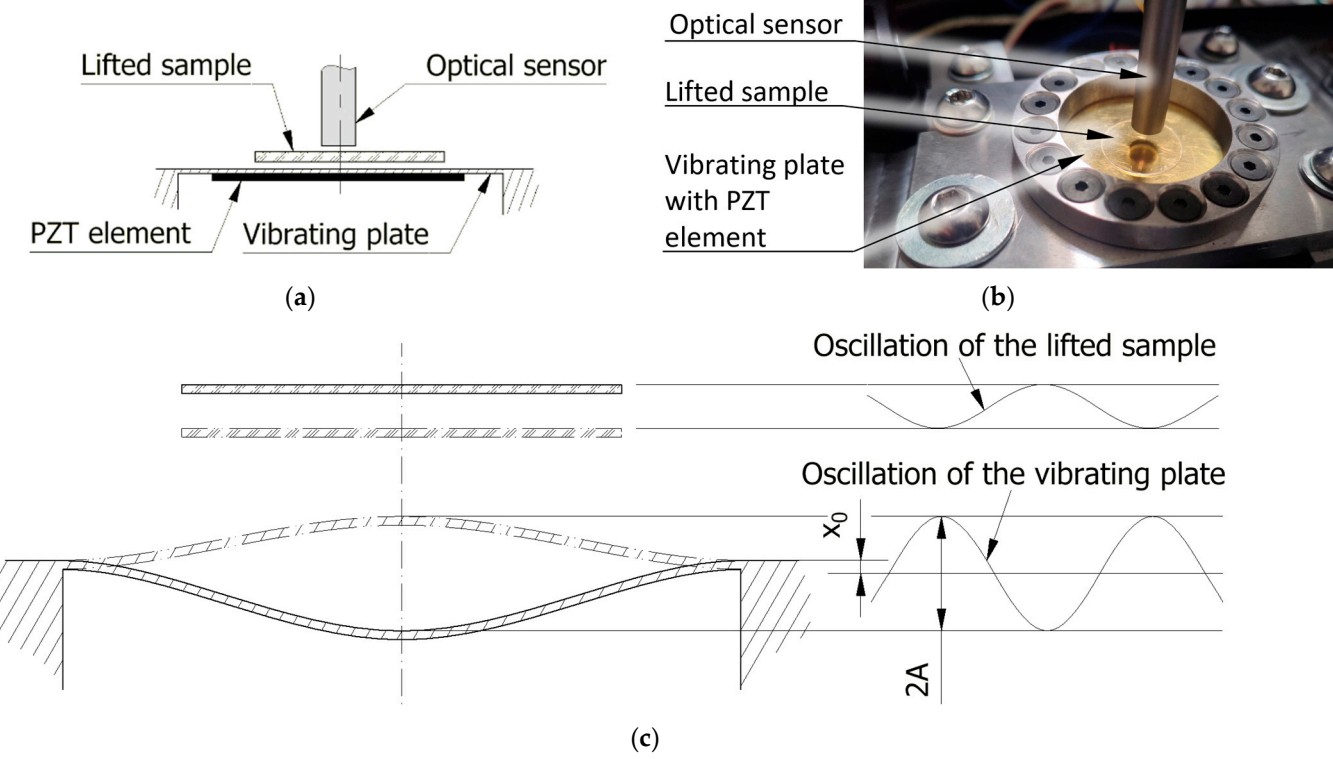

**Figure 2.** Test setup. (**a**) schematic view. (**b**) picture of the setup. (**c**) idea of the squeeze film lifting effect, where A is an oscillator vibration amplitude [*m*], and $x_0$ is a mean deflection of the oscillator [*m*].

The prototype of the plate was designed as a solid block in which a cavity was machined to form a circular membrane with a diameter of 40 mm and a thickness of 1 mm (Figure 3). The thickness of the membrane was limited due to the geometrical presumption of the plate flatness and observing manufacturing technique limitations. After the primary test, the plate has not presented satisfactory results, i.e., the levitation of the samples was not achieved, and the plate did not show the expected deflection during its operation. Due to the aforementioned, the decision was made to subject the plate to electro-drilling to thin the membrane. The process allowed decreasing thickness to 0.5 mm, with the undesired deterioration of the working surface flatness. The surface of the membrane had an eccentrically centrally positioned convexity of 50 μm (Figure 3d).

The thinning of the plate decreased its stiffness eightfold from $2480\frac{N}{mm}$ for plate thickness of 1 mm to $308\frac{N}{mm}$ for 0.5 mm membranes thickness and allowed us to observe its vibration. The first mode of vibration, according to the performed FEM calculations, should occur at 4583 Hz; the obtained experimental results showed a value of 4260 Hz. The inconsistencies can be attributed to the limited knowledge about the mass distribution of the PZT element and the cohesive agent. Moreover, in the calculations, some elements, such as the power supply connectors, are not taken into consideration. The levitation height measured for a Ø40 sample of a mass of 1.2 g was 9 μm. The height of the levitation was lower than the existing inaccuracies on the surface; therefore, the researchers decided that

an alternative design had to be applied to create a working surface of improved flatness (planarity).

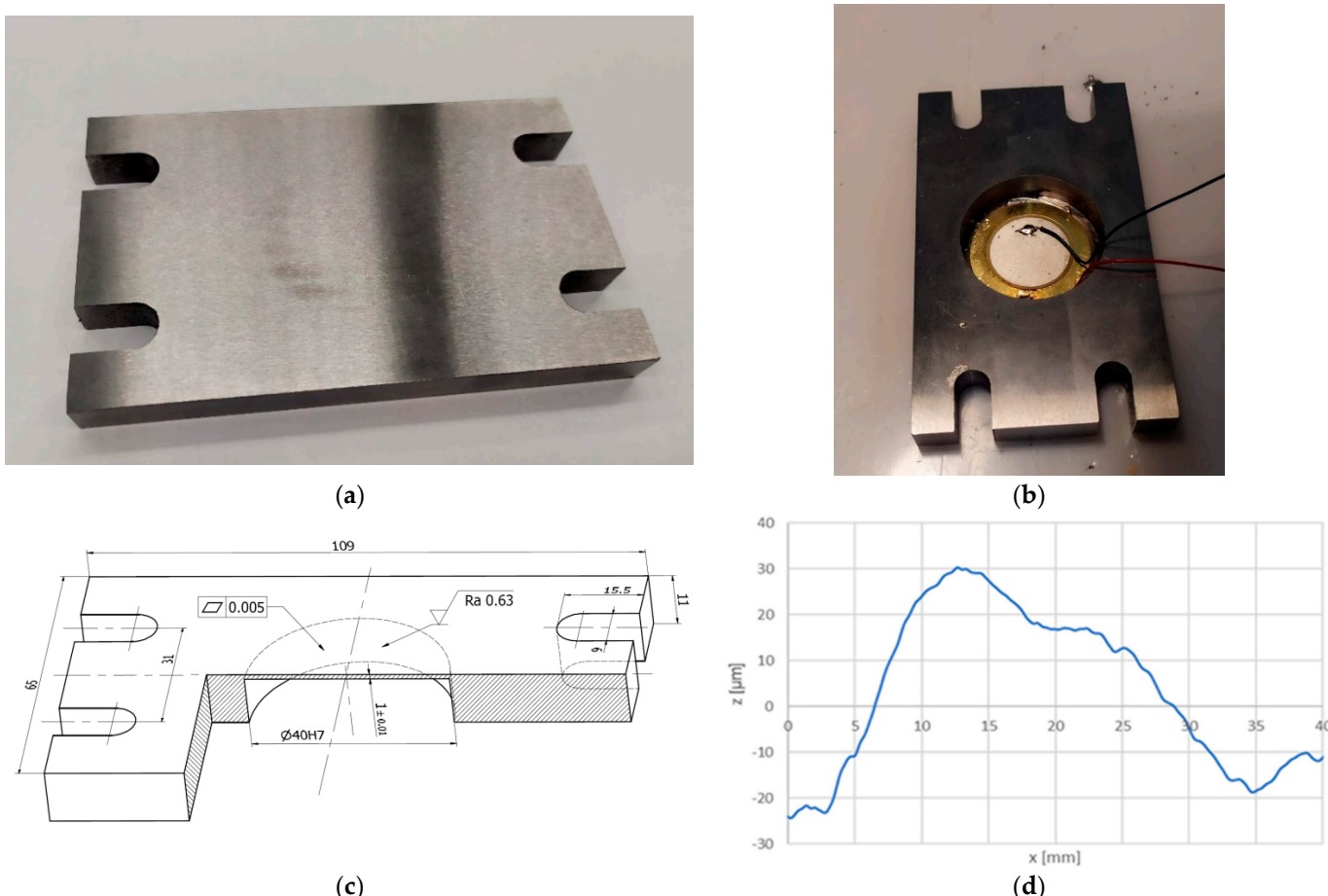

**Figure 3.** Prototype of the vibrating plate—solid block (**a**) view on the working surface. (**b**) view on the cavity with the PZT element. (**c**) geometry of the plate. (**d**) profile of the solid plate after electro-drilling.

The alternative design setup (Figure 4) is composed of the base plate (1) and comprises a membrane (2) and a thrust ring (3). The base plate acts as a rigid fixing, while the thrust ring secures the membrane. Such a design allows for a simple exchange of the membrane. The membrane used primarily was the membrane of calibrated steel foil (0.5 mm thick) (Figure 4a) and the brass PZT element (0.3 mm thick) (Figure 4c).

There were nine samples used in the experiment and verification calculations. Circular microscope glass cover slides were used as samples. Their masses and sizes are presented in Table 1. Three sets of sample diameters were used; in each set, three samples with different masses were prepared. The samples were prepared by creating stacks of glass circles using adhesive and creating samples of doubled and tripled mass.

**Table 1.** The masses of the samples used in the experiment.

| Sample Diameter | φ20 | φ25 | φ30 |
|---|---|---|---|
| | | Mass of Samples [g] | |
| Single sample (×1) | 0.118 | 0.177 | 0.270 |
| Double sample (×2) | 0.236 | 0.390 | 0.511 |
| Triple sample (×3) | 0.373 | 0.560 | 0.799 |

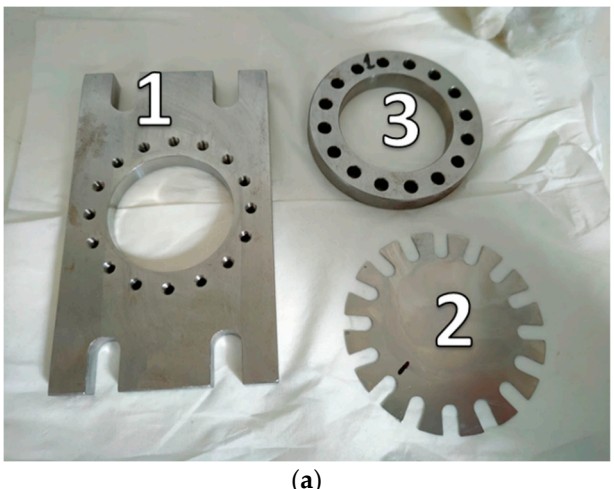
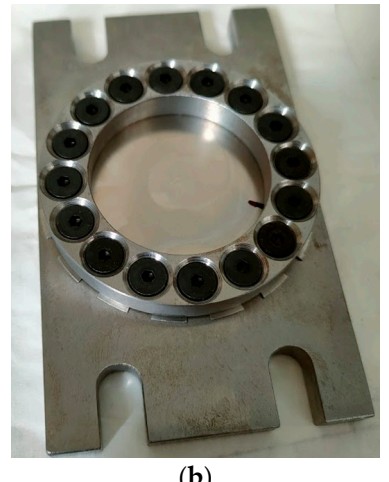
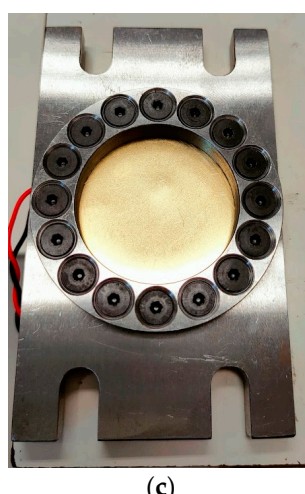

(a)
(b)
(c)

**Figure 4.** Prototype of the membrane with thrust ring: (**a**) disassembled; (**b**) assembled with a steel membrane; (**c**) assembled with PZT element.

### 2.2. Description of the CFD Model

Calculations were conducted using commercially available software [25]. The model describes the air film between the membrane and the lifted object. Due to software limitations, a circular sector of the film was utilized instead of a 2D grid. Due to the symmetry of the phenomenon, a cyclical model was adapted. Consequently, the movement of the lifted object was reduced to vertical motion above the membrane, preventing any horizontal movement of the object. The utilized sector had a central angle of $\alpha = 1°$, an initial height of $H = 7$ μm, and a radius equivalent to that of the modeled scenario. The authors used a structural mesh; mesh independence was conducted, and an exemplary result of maximum pressure is presented in Figure 5. The mesh has been divided into 450 finite volumes in the radial direction and 18 in the film thickness direction. The time step was set based on the mesh independence study as 1/120 part of one cycle of inducer motion. The scheme of the model division is presented in Figure 6. Sym1 and Sym2 regions in the model describe cyclic symmetry faces. The bottom wall describes the deflecting surface of the membrane. The deflection geometry was realized using two modes previously described.

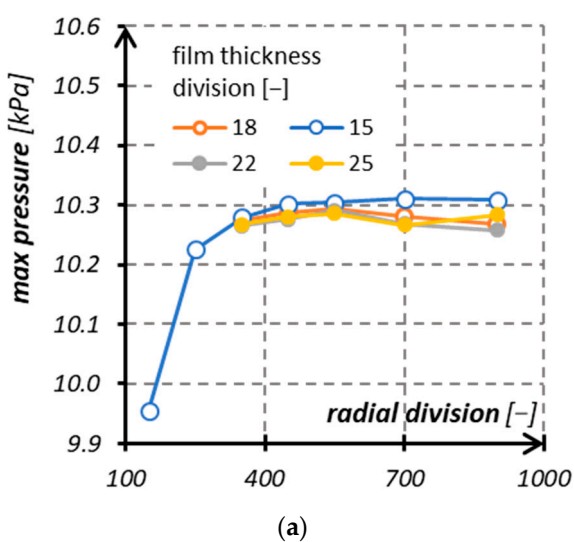
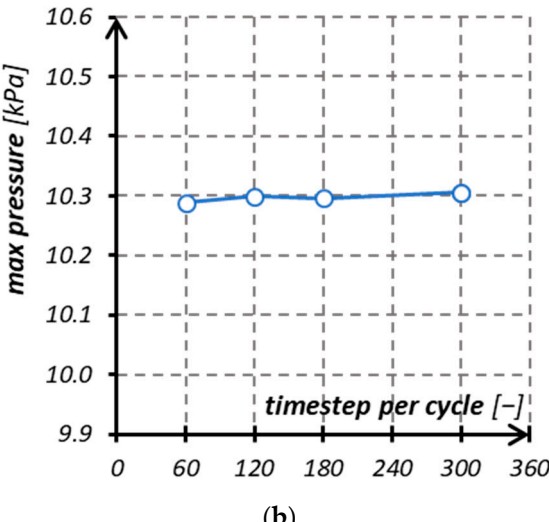

(a)
(b)

**Figure 5.** Peak pressure observed in the film over a single cycle: (**a**) mesh independence study; (**b**) timestep independence study.

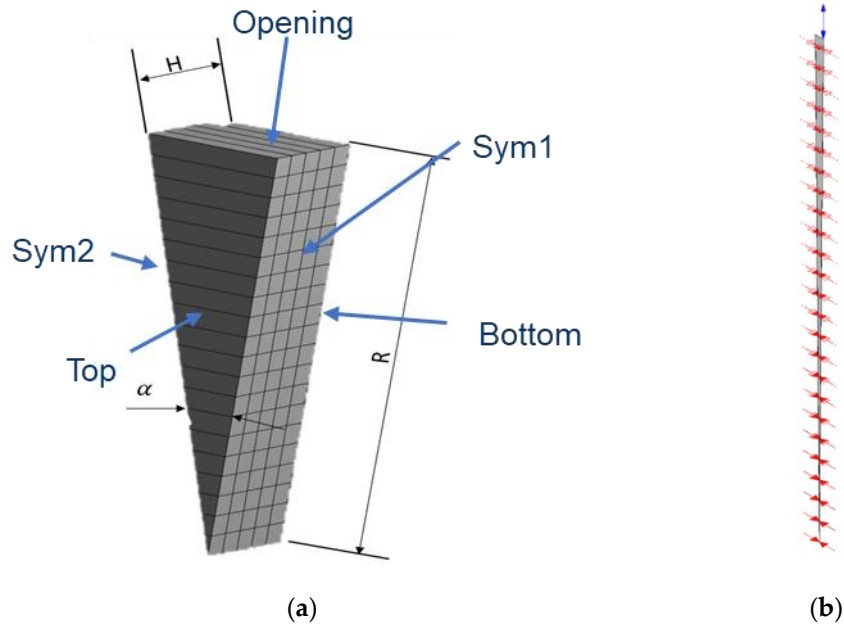

**Figure 6.** Grid of the FVM of the film: (**a**) idea of the division; (**b**) model drawn in real proportions between angle and radius.

The surface of the lifted object in the model is represented as a Rigid Body and is represented by the Top surface. The object's kinematics is obtained by solving the motion equation. The previous assumption of symmetry allows for 1DOF motion described by the equation of motion (1).

$$m\ddot{z} = F_{aero} - mg \tag{1}$$

where $m$ is the mass of the lifted object; $\ddot{z}$ is the acceleration of the lifted object, $F_{aero}$ are the sum of the forces due to the air gap, $g$ is a gravitational acceleration $g = 9.81$ kg $\frac{m}{s^2}$. The motion of both the membrane and the lifted objects generates a deformation of the model. With the use of the moving mesh technique, used structural mesh has been preserved. The model uses the Second Order Backward Euler transient scheme. The convergence criterion for the calculations is set by obtaining RMS of below $10^{-6}$. During the calculation, the finite volume method was adopted, solving continuity and ideal gas state equations: mass conservation Equation (2); momentum Equation (3); and ideal gas Equation (4). The tensor used in the momentum equation is given by Equation (5).

$$\frac{\partial \rho}{\partial t} + \nabla \cdot \left( \rho \vec{v} \right) = 0 \tag{2}$$

$$\frac{\partial \left( \rho \vec{v} \right)}{\partial t} + \nabla \cdot \left( \rho \vec{v} \times \vec{v} \right) = -\nabla p + \nabla \cdot \tau \tag{3}$$

$$\rho = \frac{Mp}{RT} \tag{4}$$

$$\tau = \mu \left( \nabla \vec{v} + \left( \nabla \vec{v} \right)^T - \frac{2}{3} \delta \nabla \cdot \vec{v} \right) \tag{5}$$

where the Identity matrix is designated $\delta$. In Equations (2)–(5), the mathematical operators are as follows: $\nabla$—divergence; $\vec{v} \times \vec{v}$ is the dyadic operator of velocity vectors $\vec{v}$; $\nabla$ is gradient; $\rho$—density of the fluid; $p$—pressure; $\tau$—stress tensor; $M$—gas molar mas; $R$—gas constant $R = 8.31446 \frac{J}{K \cdot mol}$; $T$—temperature; $\vec{v}$—velocity vector; $\mu$—dynamic viscosity.

## 3. Results

### 3.1. Verification of the CFD Model

Initially, the behavior of the vibration plate of the experimental setup was tested. For the metal sheet membrane, the first mode of vibration was identified at f = 2.76 kHz. The position of the center of the plate is presented in Figure 7. The deflection of the plate is non-symmetrical due to asymmetrical voltage parameters supplied to the PZT element. The mean value of deflection is determined at $x_0$ = −5 μm with the amplitude of A = 35 μm.

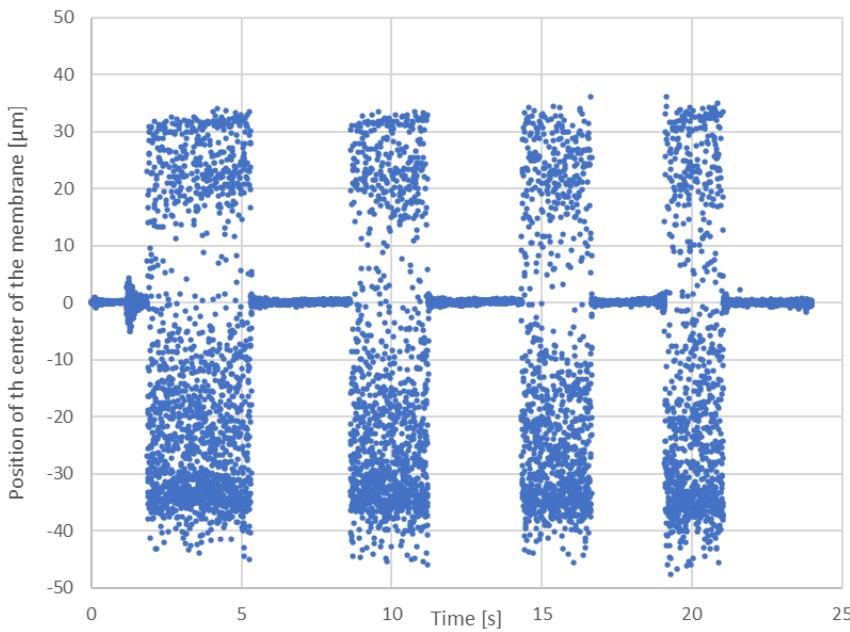

**Figure 7.** Deflection of the vibration plate at the first mode of vibration.

The results in Figure 8 show the height of levitation of the sample positioned above the membrane. The value of the levitation height through the test was equal to 140 μm, with the amplitude of vibration at 20 μm.

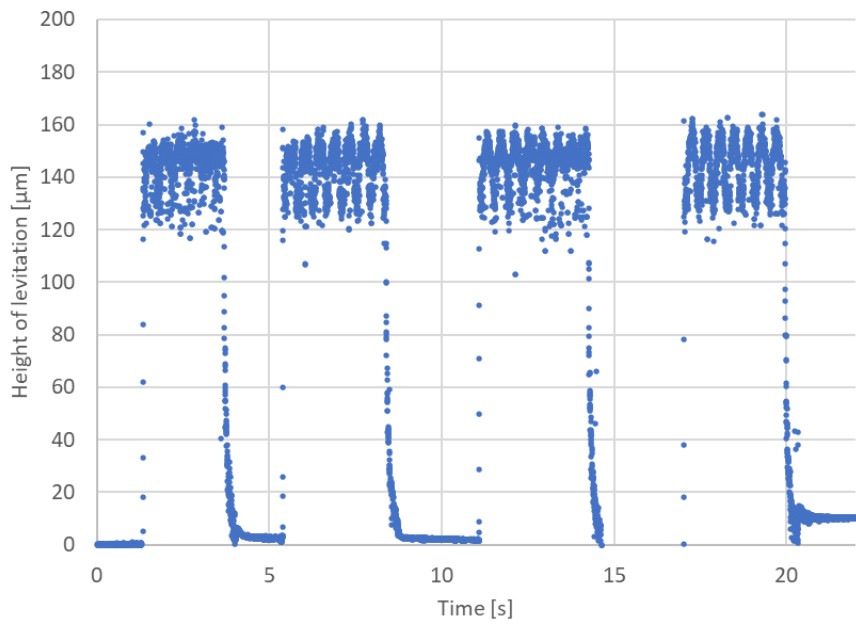

**Figure 8.** The experimental results of the lifting of the sample (ϕ20 mm, m = 0.373 g, f = 2.76 kHz).

A part of the profile presented was used as input for the fluid film calculation. The result for the single profile is presented in Figure 9. The value of the mean amplitude of the vibration is equal to 140 μm, which directly corresponds to the experimental value. The value of the amplitude differs significantly. The CFD results are equal to approximately 10 μm, whereas the experimental value is read to be twice thereof. The discrepancies can be attributed to the limited capacity of the model. The model is unable to describe the lateral motion of the sample, and similarly, there is a lack of capacity to describe the non-parallel motion (e.g., precession) of the sample.

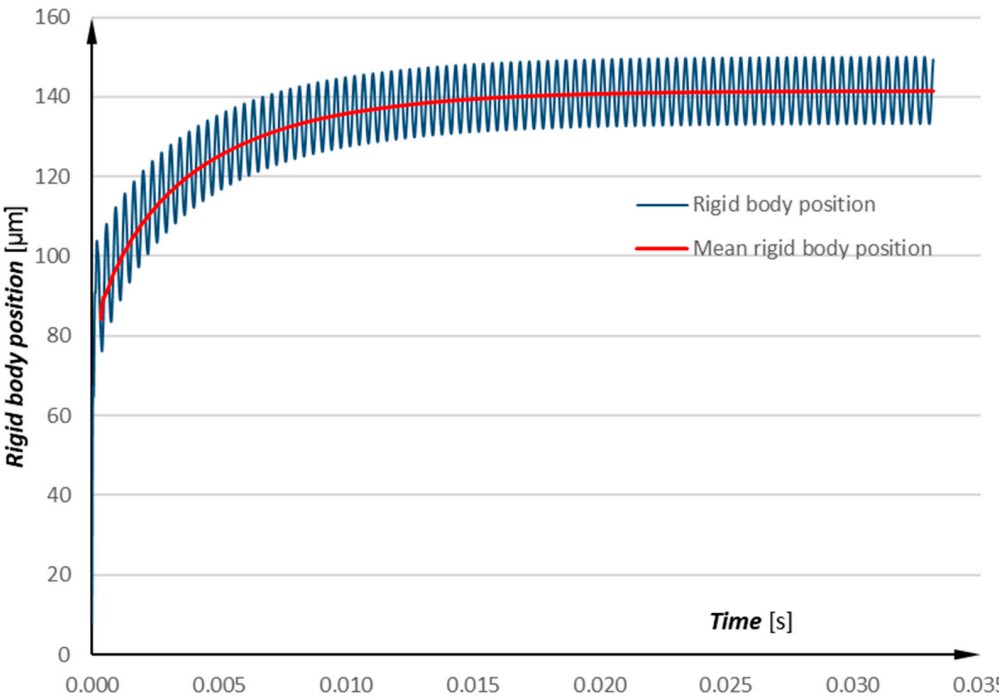

**Figure 9.** CFD calculation results for the sample (ϕ20 mm, m = 0.373 g, f = 2.76 kHz).

### 3.2. Results of Spherical Deformation

The series of CFD calculations that were carried out assumed the idealized shape of the oscillator (Figure 10). The surface was modeled as an ideally spherical dimple with a varying diameter of the sphere and a constant position of the edge fixed at ϕ40. Frequencies of the bottom surface were $f = 5$, 10, 15, 20 kHz. The shape of the dimple was modeled as perpetually (concaved, with controlled amplitude $A$ and the mean position of the center of oscillator $x_0 = 0.5$ μm $+ A$. The samples used for the calculations were modeled with masses between 5 g and 20 g with 5 g increments and a diameter of ϕ40.

The example of lifting results is shown in Figure 11. The figure shows the relation between the movement of the sample and the oscillator. The motion of the sample is characterized by the same frequency as the motion of the oscillator, which was observed for all inspected cases. Additionally, the motion of the lifted object and the membrane are generally in counter-phase.

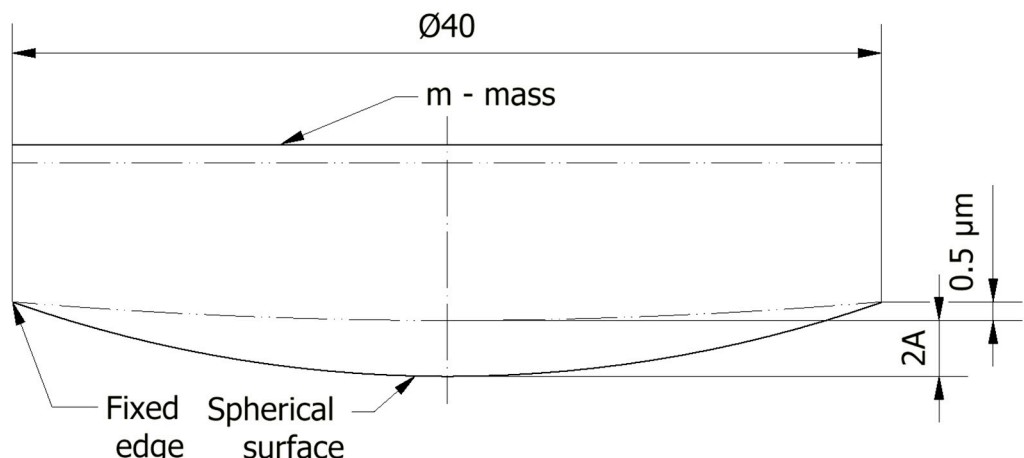

**Figure 10.** Idea of the geometry of spherical deformation calculation.

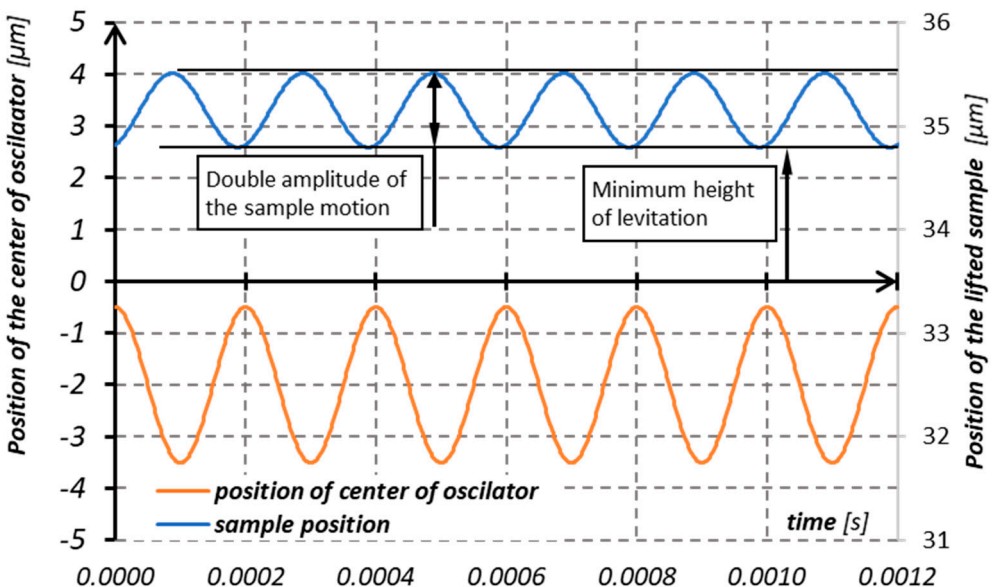

**Figure 11.** Example of motion of the sample in relation to the motion of the oscillator $m = 10$ g; $f = 5$ kHz; $A = 1.5$ µm.

Figure 12 shows the CFD results of minimal lifting heights of the masses over the spherically deformed dimple. In all cases, there is a clear decline in the levitation height while the frequency of kinematic excitation of the membrane is raised. The levitation height of the sample asymptotically decreases. In the results, the most beneficial heights are at the lowest frequencies (i.e., 5 kHz). The increase of the height is restricted by the lift-off conditions—frequency and amplitude of the vibrating plate—the phenomenon described by Brunetiere et al. [12]. The lifting height also decreases with the decreasing amplitude of the vibration of the membrane and the increase of the mass of the object lifted. The influence of the membrane motion magnitude can be attributed to the amount of energy added to the system with each motion cycle; the larger amplitude of motion of the membrane, the higher lifting heights can be achieved. The heights achieved in the calculations vary from 5 to 60 µm. The heights achieved in this experiment are higher than those reported by other researchers.

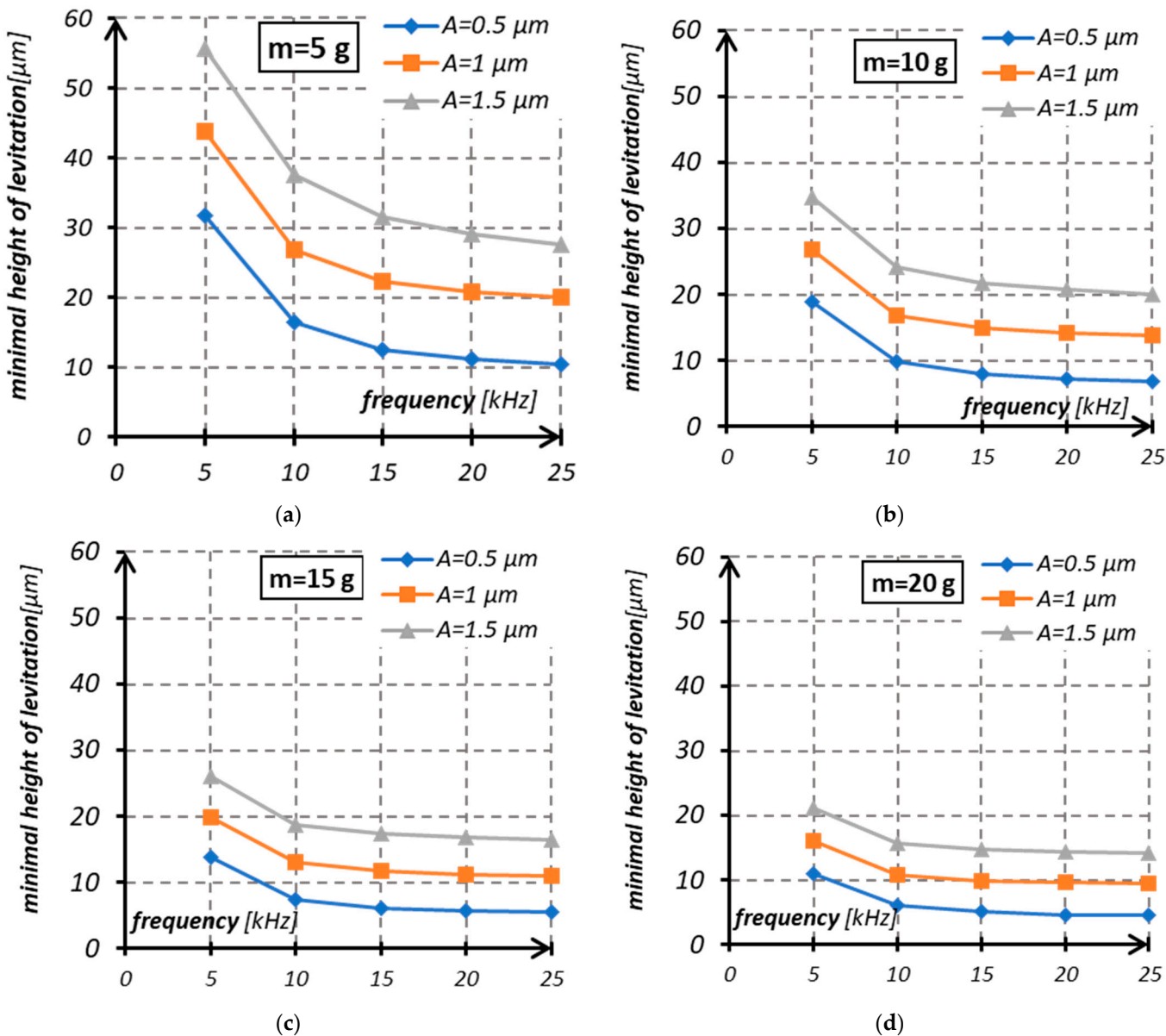

**Figure 12.** The minimal levitation height based on the CFD model of spherically deflecting membrane for masses: (**a**) 5 g; (**b**) 10 g; (**c**) 15 g; (**d**) 20 g.

The behavior of the modelled sample during the levitation is commonly overlooked during the experimental and theoretical results. The analysis of the results of the motion of the lifted elements shows a radical decrease in the amplitude of lifted objects with the increase in the frequency of excitation (Figure 13). The major factor governing the dependency is the inertia of the objects. In systems characterized by a higher frequency, the motion energy emitted by the vibrating system is dampened by the fluid film and not transferred to the object itself. It is worth noting that in all cases, the amplitude of motion of the lifted object is always lower than that of the excited plate.

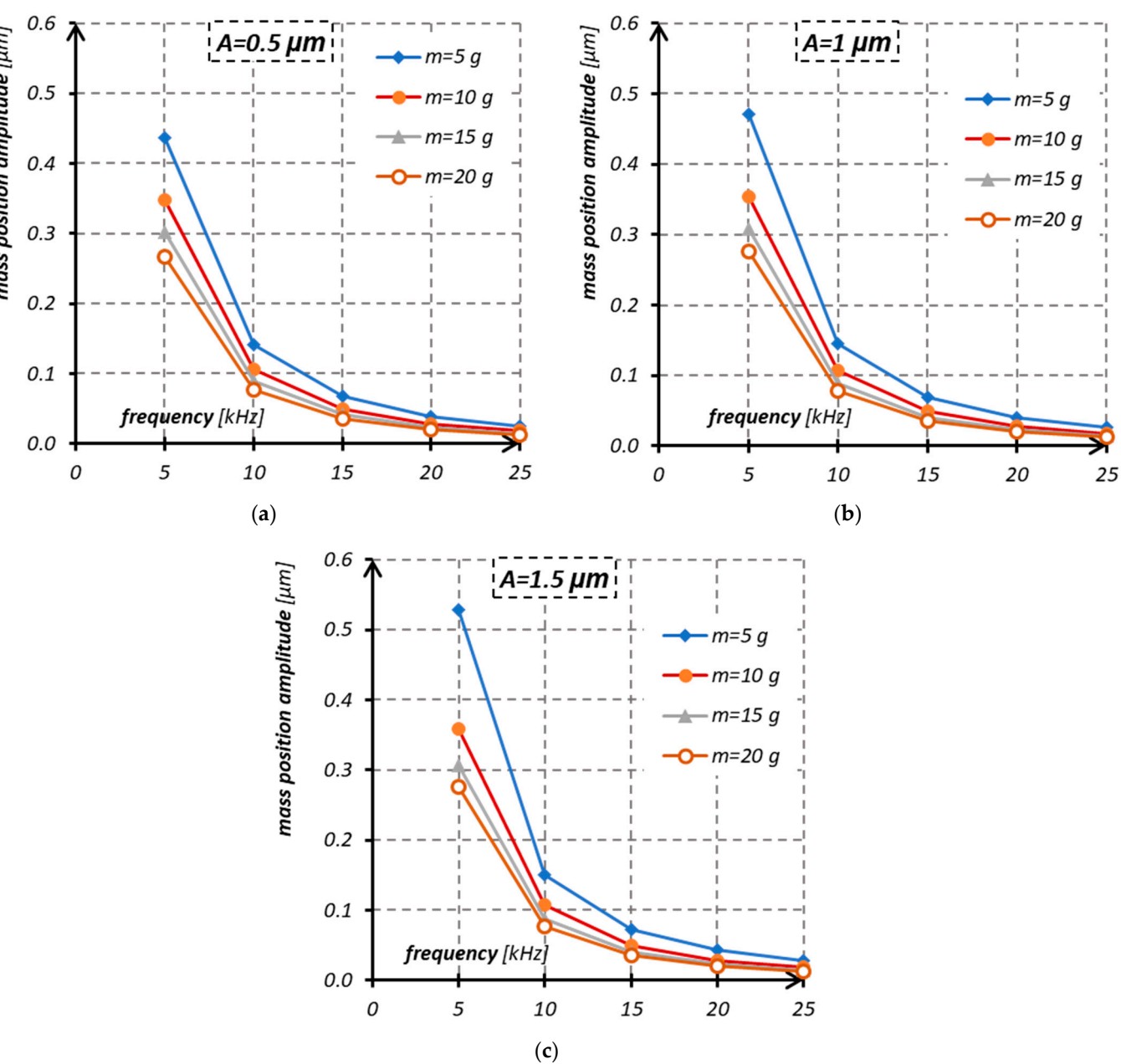

**Figure 13.** The amplitude of the modelled sample motion for excitation of the membrane of (**a**) 0.5 μm, (**b**) 1 μm, and (**c**) 1.5 μm.

Pressure profile changes in progressive time steps are presented in Figure 14 for two separate cases, which differ from the frequency of oscillations. Pressure profiles for low frequency (5 kHz) are more complex and irregular than those calculated at higher frequency (25 kHz). One can notice that a similar level of absolute pressure values was calculated irrespective of the analyzed frequency; however, cases with net positive pressures have slightly higher values than those with negative pressures. This allows to generate a small positive pressure that allows the lifting of the object. Velocity contours for the presented cases (Figure 15) are shown for the cross-section of the fluid film (the left-hand side is an axis; the right-hand side is an opening); for purposes of the figure, the fluid film has been exaggerated 50 times. The contours show more significant differences between the cases. Maximal fluid velocity for higher frequency case is significantly lower (2.72 m/s for 5 kHz, 0.58 m/s for 25 kHz), which is evidence of the limited airflow through the gap at high oscillation frequency.

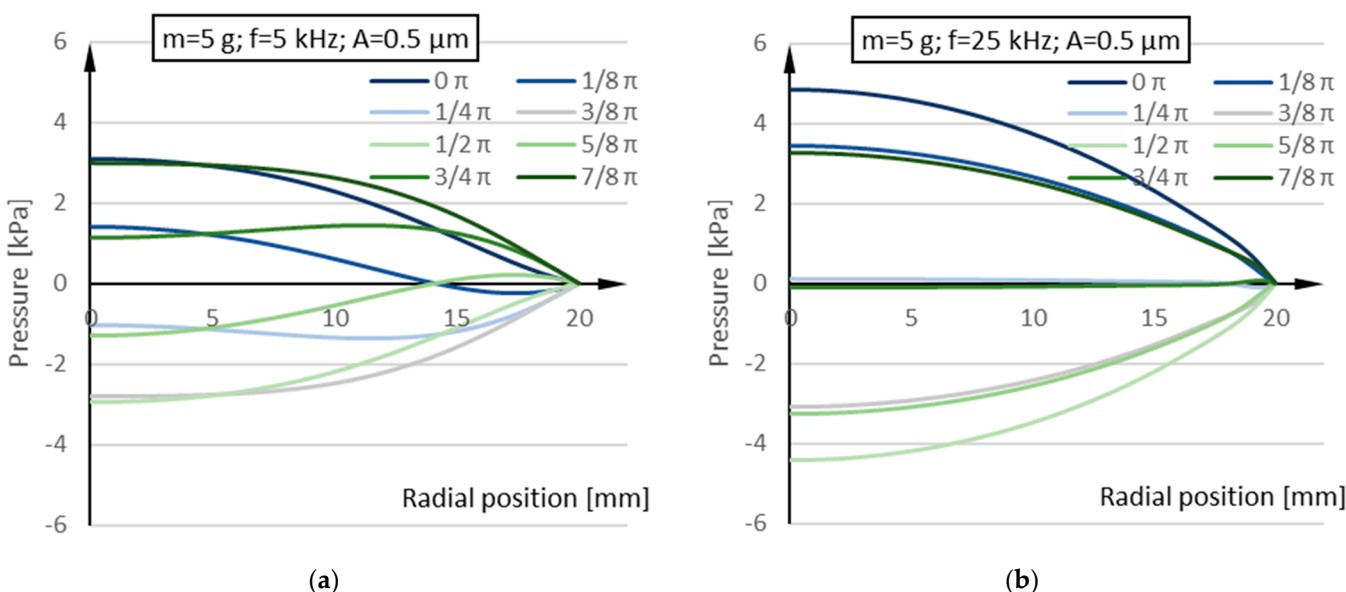

**Figure 14.** Radial pressure profiles for cases (**a**) m = 5 g; f = 5 kHz; A = 0.5 μm; (**b**) m = 5 g; f = 25 kHz; A = 0.5 μm.

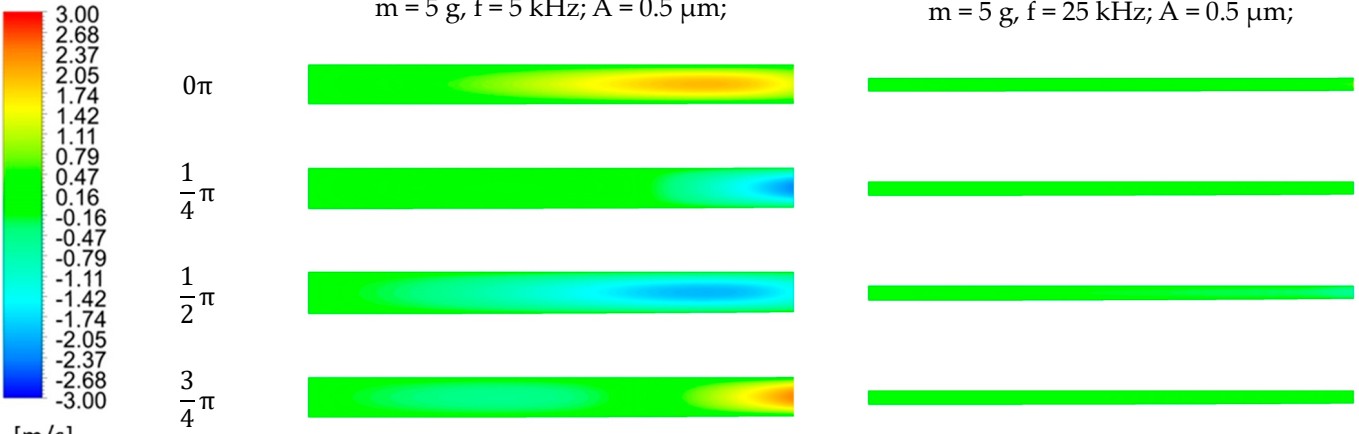

**Figure 15.** Fluid radial velocity contours for cases m = 5 g; f = 5 kHz; A = 0.5 μm and m = 5 g; f = 25 kHz; A = 0.5 μm (air thickness increased proportionally ×50 to improve the readability of the drawing).

The higher frequency forces the use of the Navier–Stokes equations rather than simplified Reynolds equations. The reasoning is described by Brunetiere et al. [19]. Authors proved that using a criterion of Helmholtz number, i.e., a ratio of time of travel of acoustic wave through the film by oscillation period. For high-frequency cases, the solutions of the Reynolds equation were proven to be inaccurate.

### 3.3. Experimental Results

The results of the levitation heights are presented in Figure 16. The measurements were carried out at 1.31 kHz with differing excitation amplitudes: 6.2; 17.1; 38.3; 57.1 μm. The parameters of the samples are shown in Table 1. The experimental results show similar behavior as predicted by calculations. The levitation height rises with the rising amplitude of motion for the oscillator. The increase of the mass of the lifted sample with the constant area of the load proves to decrease the obtained levitation height.

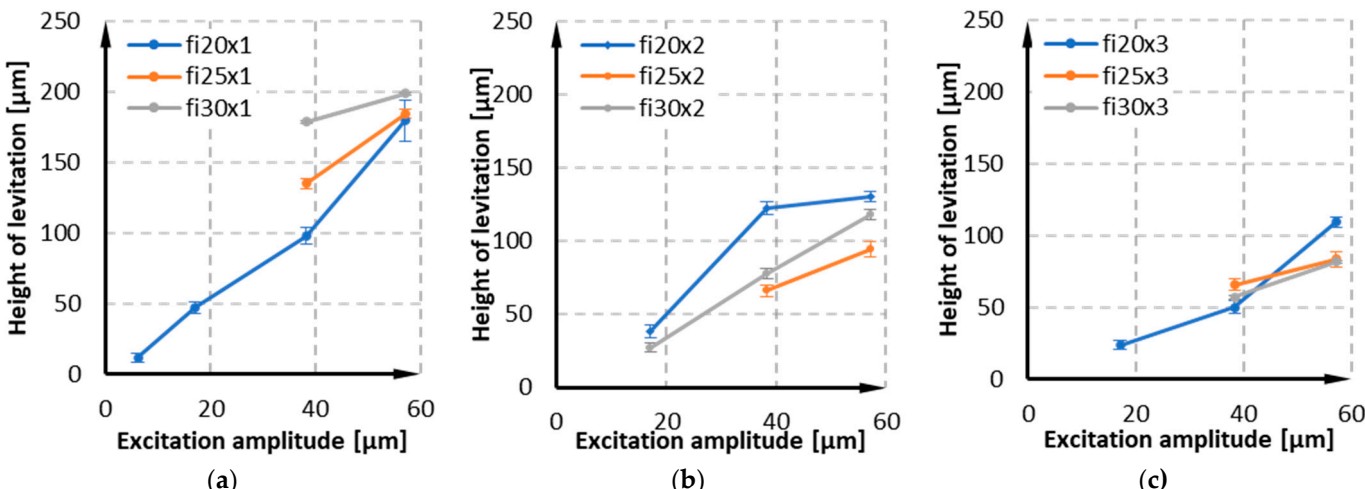

**Figure 16.** Height of levitation of samples: (**a**) single; (**b**) double stack; (**c**) triple stack.

With differing masses and diameters of the lifted object, there is an opposite influence of mass increasing with the diameter and causing the decrease in the levitation height and, on the other hand, the influence of increasing area of the sample, which undoubtedly causes the increase of levitation height. To show these effects, at the same time, the levitation heights are shown as a function of mean pressure in the gas film, being the quotient of the weight of the samples in [N] and the area in [m$^2$]. This result is shown in Figure 17, and it again shows the dependence of the heights on the amplitude.

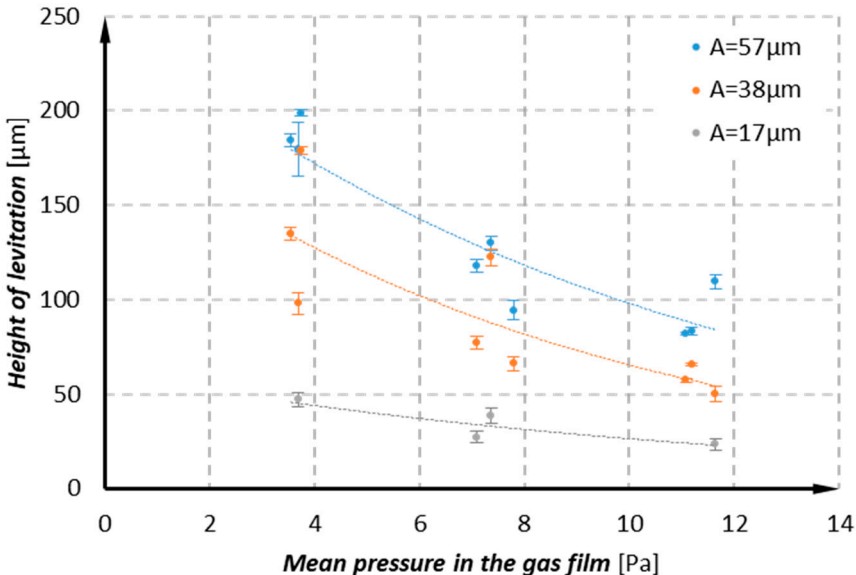

**Figure 17.** Levitation height [μm] as a function of mean pressure in the gas film [Pa] for various amplitudes of the membrane [μm].

## 4. Conclusions

The model developed and used by the authors allows for the observation of not only the height of the levitation but also allows observing the motion of the sample itself.

The model was verified by an experiment with a very good agreement of the levitation height and worse prediction of the object oscillations during levitation.

The use of the Navier–Stokes equations, as opposed to the Reynolds equations, provides more reliable results. It allows for detailed observations of the fluid film behavior

and, in the future, may be used for creating a full FSI model with a flexible sample and multiple actuators.

There is a clear influence of the mass on the lifting height—the higher load causes a thinner film. The influence of the frequency is shown, and the increase of the frequency causes an asymptotic decrease in height.

The amplitude of motion of the sample was observed. It was determined that the amplitude of the motion of the sample in stationary conditions is smaller than the amplitude of the excitation plate (membrane). Moreover, the motion of the sample and the membrane is in counter-phase. This is important in case of small levitation heights, as there is a possibility of contact between the sample and membrane, with a risk of destruction in case of vulnerable elements.

**Author Contributions:** Conceptualization, B.B., R.G. and M.W. (Michał Wodtke); methodology, B.B., R.G.; software, M.W. (Michał Wodtke) and B.B.; validation, B.B., R.G. and M.W. (Michał Wasilczuk); formal analysis, B.B.; investigation, B.B., R.G.; resources, M.W. (Michał Wasilczuk); data curation, B.B.; writing—original draft preparation, B.B.; writing—review and editing, M.W. (Michał Wasilczuk) and R.G.; visualization, B.B. and R.G.; supervision, M.W. (Michał Wasilczuk); project administration, M.W. (Michał Wasilczuk); funding acquisition, M.W. (Michał Wasilczuk). All authors have read and agreed to the published version of the manuscript.

**Funding:** This research was funded by Polish National Science Centre, grant number 2016/23/B/ST8/00210.

**Institutional Review Board Statement:** Not applicable.

**Informed Consent Statement:** Not applicable.

**Data Availability Statement:** The data presented in this study are available from the authors upon reasonable request.

**Acknowledgments:** CFD computations were carried out using the computers of the Centre of Informatics Tricity Academic Supercomputer and Network (TASK) in Gdansk.

**Conflicts of Interest:** The authors declare no conflict of interest.

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
