# Peer review of "Experimental Verification of the CFD Model of the Squeeze Film Lifting Effect"

_applsci, doi:10.3390/app13116441_

Round 1

Reviewer 1 Report

the authors presented and experimental and numerical study of the squeeze film lifting effect.

The main quantitative findings are to be mentioned in the abstract.

The novelty of the paper is to be clearly stated.

An actual photo of the experimental setup is to be presented.

Details about the measurement techniques and data acquisition system are to be provided.

An experimental uncertainty study is to be performed.

Figures 2, 3, 8, 9 and 14  have low resolution.

The numerical method is to be detailed.

What is the used numerical method? In the title of Fig 5, it is FEM but in line 200 it is FVM.

What is the used software?

Based on Fig 5, the used mesh seems to be coarse. A grid sensitivity test is to be performed.

Line 201, is to be checked.

The boundary conditions are to be expressed mathematically.

What is the convergence criterion?

What is the used time step?

The authors studied a 3D configuration without presenting any 3D profile.

Results related to flow structure are to be added and discussed.

The paper is to be checked against misprints and grammatical mistakes.

The paper is to be checked against misprints and grammatical mistakes.

Author Response

The authors would like to thank Reviewer 1 for his detailed assessment and helpful comments. We addressed all your remarks and introduced changes to our submission

The main quantitative findings are to be mentioned in the abstract.

Response: Most relevant findings were added.

The novelty of the paper is to be clearly stated.

Response: Added statement about the novelty

An actual photo of the experimental setup is to be presented.

Response: Picture 2 b is added – it shows the experimental setup.

Details about the measurement techniques and data acquisition system are to be provided.

Response: Information about the equipment used was added.

An experimental uncertainty study is to be performed.

Response: The error bars are added to the graphs with experimental results (Fig. 15 and 16)

Figures 2, 3, 8, 9 and 14  have low resolution.

Response: Corrected – Fig 9 (Fig. 8 in the  original submission) changed

The numerical method is to be detailed.

What is the used numerical method? In the title of Fig 5, it is FEM but in line 200 it is FVM.

Response: FVM is used - the title of the figure was previously misleading

What is the used software?

Response: Statement about the used software was added.

Based on Fig 5, the used mesh seems to be coarse. A grid sensitivity test is to be performed.

Response: The figure illustrates the concept of division (a) but not the actual proportions of the shape and mesh density. The proper proportions are showed on the second figure (b).

The mesh independence study and information about the mesh density has been added  (Fig. 5).

Line 201, is to be checked.

Response: The mistake of incorrect reference was corrected

The boundary conditions are to be expressed mathematically.

Response: these are very basic boundary conditions and we did not think it would be helpful to add the formula for them

What is the convergence criterion?

Response: Convergence criterion for the calculations is set by obtaining RMS of below 10-6 – information added.

What is the used time step?

Response: Information about the time step was added (also timestep independence study in Fig. 5b)

The authors studied a 3D configuration without presenting any 3D profile.

Response: The 3D model was used because of the software limitations - but is a cyclical 2D model.

Results related to flow structure are to be added and discussed.

This was not our intention in the first paper on this research, but in the revision we added Figure 14 with some more detailed CFD study results.

The paper is to be checked against misprints and grammatical mistakes.

The paper was once more checked

Comments and Suggestions for Authors

the authors presented and experimental and numerical study of the squeeze film lifting effect.

The main quantitative findings are to be mentioned in the abstract.

Response: Most relevant findings were added.

The novelty of the paper is to be clearly stated.

Response: Added statement about the novelty

An actual photo of the experimental setup is to be presented.

Response: Picture 2 b is added – it shows the experimental setup.

Details about the measurement techniques and data acquisition system are to be provided.

Response: Information about the equipment used was added.

An experimental uncertainty study is to be performed.

Response: The error bars are added to the graphs with experimental results (Fig. 15 and 16)

Figures 2, 3, 8, 9 and 14  have low resolution.

Response: Corrected – Fig 14 (Fig 9 in the  original submission) changed

The numerical method is to be detailed.

What is the used numerical method? In the title of Fig 5, it is FEM but in line 200 it is FVM.

Response: FVM is used - the title of the figure was previously misleading

What is the used software?

Response: Statement about the used software was added.

Based on Fig 5, the used mesh seems to be coarse. A grid sensitivity test is to be performed.

Response: The figure illustrates the concept of division (a) but not the actual proportions of the shape and mesh density. The proper proportions are showed on the second figure (b).

The mesh independence study and information about the mesh density has been added  (Fig. 5).

Line 201, is to be checked.

Response: The mistake of incorrect reference was corrected

The boundary conditions are to be expressed mathematically.

Response: these are very basic boundary conditions and we did not think it would be helpful to add the formula for them

What is the convergence criterion?

Response: Convergence criterion for the calculations is set by obtaining RMS of below 10-6 – information added.

What is the used time step?

Response: Information about the time step was added (also timestep independence study in Fig. 5b)

The authors studied a 3D configuration without presenting any 3D profile.

Response: The 3D model was used because of the software limitations - but is a cyclical 2D model.

Results related to flow structure are to be added and discussed.

This was not our intention in the first paper on this research, but in the revision, we added Figure 14 with some more detailed CFD study results.

The paper is to be checked against misprints and grammatical mistakes.

The paper was once more checked

Reviewer 2 Report

The authors have attempted to build CFD model for prediction of levitation height using Near Field acoustics levitation method. Even though novelty of the work was presented well, technical details of computational model are seriously missing. The authors have to take into consideration of the comments below, before the manuscript can be accepted,

1. The abstract has to be rewritten to include 2-3 lines about the major observations of the work.

2. The section 2.2 which describes the the CFD model is quite incomplete like

i) what the grid side using in this work

ii) what is computational domain used. The details given in figure 5 are not complete

iii) what was time step size

iv) Did authors looked whether chosen time step gives Time step independent solution

v) Did the authors used commercial code or code developed in the group?

vi) how did the authors modelled the deflection of the bottom of the computational domain

vii) what is the stiffness of the vibration plate?

3. The authors are presenting the levitation height as results. The reasoning for the trends observed have to explain further by using velocity and pressure contours. Then only there is purpose for development of detailed CFD mdoel

The authors writing was easy to understand. 

Author Response

  1. The abstract has to be rewritten to include 2-3 lines about the major observations of the work.

Response: Most important values obtained in the course of the research was added to the abstract.

  1. The section 2.2 which describes the the CFD model is quite incomplete like
  2. i) what the grid size using in this work

Response: Grid size with mesh independence study were added

  1. ii) what is computational domain used. The details given in figure 5 are not complete

Response: The domain details were added.

iii) what was time step size

Response: Time step information was added (1/120 of the inducer motion cycle)

  1. iv) Did authors looked whether chosen time step gives Time step independent solution

Response: Timestep independence study was added.

  1. v) Did the authors used commercial code or code developed in the group?

Response: Statement about the software was added

  1. vi) how did the authors modelled the deflection of the bottom of the computational domain

Response: The deflection was defined as a function of position in function of time. Moving mesh function was used so that the mesh structure was preserved.

vii) what is the stiffness of the vibration plate?

The membrane stiffness was added

  1. The authors are presenting the levitation height as results. The reasoning for the trends observed have to explain further by using velocity and pressure contours. Then only there is purpose for development of detailed CFD model

In this paper such an analysis was omitted deliberately

Comments on the Quality of English Language

The author's writing was easy to understand. 

However we made an additional language check

Reviewer 3 Report

The general comments to the authors are as follows:

The abstract needs to be expanded and includes a brief explanation of where it can be used and some of the outcomes. Currently, the abstract only describes what the article is all about.

Line 30: what is SFAL? It appears for the first time and needs to be appropriately explained.

Line 64: Ref #6 has more than one author, so it should be First Author et al.

Line 69: Ref #7 should appear after the author's name.

The CFD section is not properly explained. The mesh needs to be explained (the element size, type of mesh, etc). Also, the mesh independency study needs to be included. Furthermore, the boundary conditions information should also be presented.

Line 213: Figure 6 appears twice.

These are some examples, the authors should review the paper again and ensure the comments are applied especially in the CFD section.

The English grammar also needs to be reviewed.

Some grammar mistake appears in the text. The authors should review the paper again.

Author Response

The authors would like to thank Reviewer 3 for his detailed assessment and helpful comments. We addressed all your remarks and introduced changes to our submission

The general comments to the authors are as follows:

The abstract needs to be expanded and includes a brief explanation of where it can be used and some of the outcomes. Currently, the abstract only describes what the article is all about.

Line 30: what is SFAL? It appears for the first time and needs to be appropriately explained.

Response: The term that should be in this sentence is SWAL not SFAL. The error was corrected

Line 64: Ref #6 has more than one author, so it should be First Author et al.

Response: The mistake was corrected.

Line 69: Ref #7 should appear after the author's name.

Response: The mistake was corrected.

The CFD section is not properly explained. The mesh needs to be explained (the element size, type of mesh, etc). Also, the mesh independency study needs to be included. Furthermore, the boundary conditions information should also be presented.

Mesh independence study and time step independence study was added.  Information about the mesh size, and specific geometry was included.

Line 213: Figure 6 appears twice.

 Response: The mistake was corrected.

These are some examples, the authors should review the paper again and ensure the comments are applied especially in the CFD section.

The English grammar also needs to be reviewed.

Comments on the Quality of English Language

Some grammar mistake appears in the text. The authors should review the paper again.

The grammar and spelling was once more checked

Round 2

Reviewer 1 Report

After revision, the paper can be accepted for publication

Author Response

The Authors would like to thank reviewer #1 for his help and his acceptance of the paper revision

Reviewer 2 Report

The authors have taken care of all the queries except comment

"The authors are presenting the levitation height as results. The reasoning for the trends observed have to explain further by using velocity and pressure contours. Then only there is purpose for development of detailed CFD model"

The response to this comments is that explanation is deliberately removed. But why?

In my opinion, this explanation is essential. 

The language is good. 

Author Response

"The authors are presenting the levitation height as results. The reasoning for the trends observed have to explain further by using velocity and pressure contours. Then only there is purpose for development of detailed CFD model"

In my opinion, this explanation is essential.

The response to this comments is that explanation is deliberately removed. But why?

In fact, the removal of this remark and comment resulted from inadequate communication between the co-authors. This remark was addressed by graphs in Figure 14 in the first revision and also Fig 15 added to the second revision, as well as explanations in the first revision and lines 299-322 in the second revision. For higher frequencies, the fluid motion is limited by the inertia effects not included in Reynolds type analysis. The reasoning for the utilization of a detailed model is described by Brunetiere et al. [1] - the authors showed that the Raynolds equation is not sufficient when the Helmholtz number is above 1 - for simulations performed the number is in the range 0,29-1,47. 

1. Brunetière, N.; Wodtke, M. Considerations about the Applicability of the Reynolds Equation for Analyzing High-Speed Near Field Levitation Phenomena. J. Sound Vib. 2020, 483, 1–13, doi:10.1016/j.jsv.2020.115496.

Reviewer 3 Report

The authors responded to all comments.

Just some mistakes in Line 199 (Figure 5 Figure 1) and 234 (Figure 7 Figure 7).

Lines 299 to 308: this paragraph needs rewriting. It is not clear.

Author Response

The Authors would like to thank reviewer#3 for his helpful comments and his acceptance of the revision of the paper.

Just some mistakes in Line 199 (Figure 5 Figure 1) and 234 (Figure 7 Figure 7) - these were generated during the export to pdf file - hopefully we corrected it

Lines 299 to 308: this paragraph needs rewriting. It is not clear

This paragraph was rewritten and also expanded a bit